# Active coacervate droplets as a model for membraneless organelles and protocells

Carsten Donau[1], Fabian Späth[1], Marilyne Sosson[1], Brigitte A. K. Kriebisch[1], Fabian Schnitter[1], Marta Tena-Solsona[1,2], Hyun-Seo Kang [1,3], Elia Salibi[4], Michael Sattler[1,3], Hannes Mutschler [4] & Job Boekhoven [1,2✉]

Membraneless organelles like stress granules are active liquid-liquid phase-separated droplets that are involved in many intracellular processes. Their active and dynamic behavior is often regulated by ATP-dependent reactions. However, how exactly membraneless organelles control their dynamic composition remains poorly understood. Herein, we present a model for membraneless organelles based on RNA-containing active coacervate droplets regulated by a fuel-driven reaction cycle. These droplets emerge when fuel is present, but decay without. Moreover, we find these droplets can transiently up-concentrate functional RNA which remains in its active folded state inside the droplets. Finally, we show that in their pathway towards decay, these droplets break apart in multiple droplet fragments. Emergence, decay, rapid exchange of building blocks, and functionality are all hallmarks of membrane-less organelles, and we believe that our work could be powerful as a model to study such organelles.

[1] Department of Chemistry, Technical University of Munich, Lichtenbergstrasse 4, 85748 Garching, Germany. [2] Institute for Advanced Study, Technical University of Munich, Lichtenbergstrasse 2a, 85748 Garching, Germany. [3] Institute of Structural Biology, Helmholtz Zentrum München, Ingolstädter Landstrasse 1, 85764 Neuherberg, Germany. [4] Max Planck Institute of Biochemistry, Am Klopferspitz 18, 82152 Martinsried, Germany. ✉email: job.boekhoven@tum.de

Membraneless organelles, which include the nucleoli and P granules, rely on liquid–liquid phase separation in order to compartmentalize biochemical reactions.[1–6] They typically comprise proteins and RNA held together by entropic and ion-pairing effects in a process called complex coacervation,[7] resulting in an organelle with liquid-like properties.[8,9] Recent work has demonstrated that some of these organelles are actively regulated by ATP-dependent chemical reaction cycles.[7,9,10] These findings suggest that some of the membraneless organelles are active droplets, i.e., these droplets exist out of equilibrium and are regulated by chemical reactions. However, how the cell regulates phase separation of these droplets remains poorly understood.

Numerous synthetic complex coacervate-based droplets exist,[11–18] which are used to investigate the characteristics of membraneless organelles.[19,20] Examples of reversible complex coacervate droplets have been reported that form or dissolve upon a change in pH,[21] temperature,[19] salt or polymer concentration,[19,22] or in response to UV light[23] or an enzymatic reaction[24–26]. These models provide valuable information on the formation or dissolution of droplets. However, these droplets evolve from one equilibrium state to another (e.g., from droplets to solution) in response to a change in their environment. An ideal model to study how dynamic assembly and disassembly affect droplet behavior would include the fuel-driven behavior as observed in biology.[27,28] Such a model could reveal insights on how chemical reactions regulate droplet growth or how kinetics can regulate droplet composition. Moreover, chemically fueled active coacervate droplets have been proposed as a protocell model capable of self-division.[29,30]

The field of fuel-driven self-assembly has recently gained traction.[31–34] Here, the self-assembly of molecules is regulated by a chemical reaction cycle that comprises two reactions: a building block activation and deactivation reaction. In the activation, a precursor molecule is activated for self-assembly by the irreversible consumption of a chemical fuel. In the deactivation, the product is reverted to the precursor. Consequently, the self-assembling product is present for a finite amount of time, and the assembly's properties are regulated by the kinetics of the reaction cycle. Such kinetic regulation is inspired by biology's mode of controlling assemblies like the membraneless organelles. Examples of synthetic dissipative assemblies include fibers driven by photochemical[35] or fuel-driven reaction cycles,[36–40] vesicles driven by ATP[41] or carbodiimides,[42] and others.[43]

In this work, we thus introduce a model for membraneless organelles based on complex coacervate droplets regulated by a fuel-driven chemical reaction cycle. We find that droplet behavior is regulated by the conversion of a chemical fuel. For example, these droplets form spontaneously and decay in the absence of fuel. Moreover, in their pathway towards decay, they show active behavior such as the formation of vacuoles and breaking apart into fragments. Finally, we show that functional RNA, like ribozymes, can be transiently up-concentrated in these droplets and can remain in its active folded state inside the droplets.

## Results and discussion
**Design of fuel-driven complex coacervation**. The active droplets are based on the complexation of RNA and a peptide. The RNA component of the droplets is homo-polymeric RNA (poly-U, ± 2,200 bases), which is frequently used in coacervate-based studies.[19,25,44,45] The droplet dynamics are regulated by a peptide of which the RNA affinity is controlled by a chemical reaction cycle. The activation reaction converts a negatively charged aspartate into its corresponding anhydride driven by the hydrolysis of EDC (1-ethyl-3-(3-dimethylaminopropyl)carbodiimide,

fuel) to its urea (EDU, Fig. 1a and Supplementary Fig. 2).[46–49] In the deactivation reaction, the anhydride product spontaneously hydrolyzes to the precursor peptide. The result is a transient anhydride at the expense of EDC.

The design criteria for the peptide were that in the anhydride state (product), it can induce complex coacervation with poly-U (RNA), but not the aspartate state (precursor). We tested several peptide designs (see Table 1), but the most successful sequence was: Ac-FRGRGRGD-OH (phenyl alanine (F), arginine (R), glycine (G), and aspartate (D), Fig. 1a). In our design, three cationic arginines were required to interact with the anionic RNA. In line by reports from others, it appeared to be crucial to separate the charges with the non-charged glycine amino acid. That motif is also a common motif in biology and membraneless organelle related proteins.[50–52] We purposefully terminated the sequence with an aspartate. Thus, in the precursor state, the overall charge of the peptide was +1, but in the product state, it was +3. In that way, droplet formation via complex coacervation is expected for the product but not the precursor (Fig. 1b).

**Transient formation of droplets**. We dissolved 23 mM of the peptide precursor in an aqueous 200 mM MES buffer solution at pH 5.3 with 4.1 mM RNA (expressed in [Uracil nucleotides]). Isothermal titration calorimetry (ITC) showed a weak interaction between the precursor and the RNA with a dissociation constant ($K_d$) of 2.9 mM (see Supplementary Fig. 3). The precursor solution was optically clear (Fig. 2a), pointing towards the absence of droplets.

In contrast, upon the addition of 25 mM EDC (fuel), the solution immediately turned turbid (Fig. 2a) as a result of droplet formation (Fig. 2b). The fuel-induced turbidity faded rapidly, and the solution was clear after 18 min. Control experiments confirmed that the transient droplet formation was a result of fuel, RNA, and peptide combined (see Supplementary Fig. 4). In addition, droplets re-emerged after the addition of a second batch of fuel (see Supplementary Fig. 5). A plate reader-study confirmed the rapid increase of turbidity after the addition of fuel to a maximum reached after 3 min (Fig. 2c, black trace). After 18 min, the turbidity decayed to its original level.

The concentrations of the product and fuel were measured by HPLC, and the data was used to develop a kinetic model that predicts the concentration of fuel, precursor, and product throughout the cycle (Fig. 2c, Supplementary Fig. 6, and Table 2). Noteworthy, one set of rate constants in our kinetic model could accurately predict the kinetic profiles of the fuel, precursor, and product. That is a good indication that the droplets do not affect the reaction kinetics of product activation or deactivation. The concentration of fuel rapidly decayed (see Supplementary Fig. 6), leading to the transient presence of product (Fig. 2c). Moreover, the time at which the fuel was depleted coincided with the time at which we detected no more product. Therefore, fuel was present throughout the cycle and, thus, activation and deactivation were always operating simultaneously. We noted a strong correlation between HPLC data and the turbidity: at the highest product concentration, the turbidity peaked, and, when the product and fuel were exhausted, the samples returned to transparent. We refer to these droplets as "dynamic droplets" since their presence and absence seemed to be dynamically regulated by the kinetics of the reaction cycle.

The droplets behaved differently when more fuel (60 mM) was added (Fig. 2d). While HPLC showed that the fuel and product were absent after 30 min (see Supplementary Fig. 6), the turbidity of the solution persisted for 76 min. The latter observation was a strong indication that these droplets were in a kinetically arrested state; despite being transient, the dynamics of disassembly were

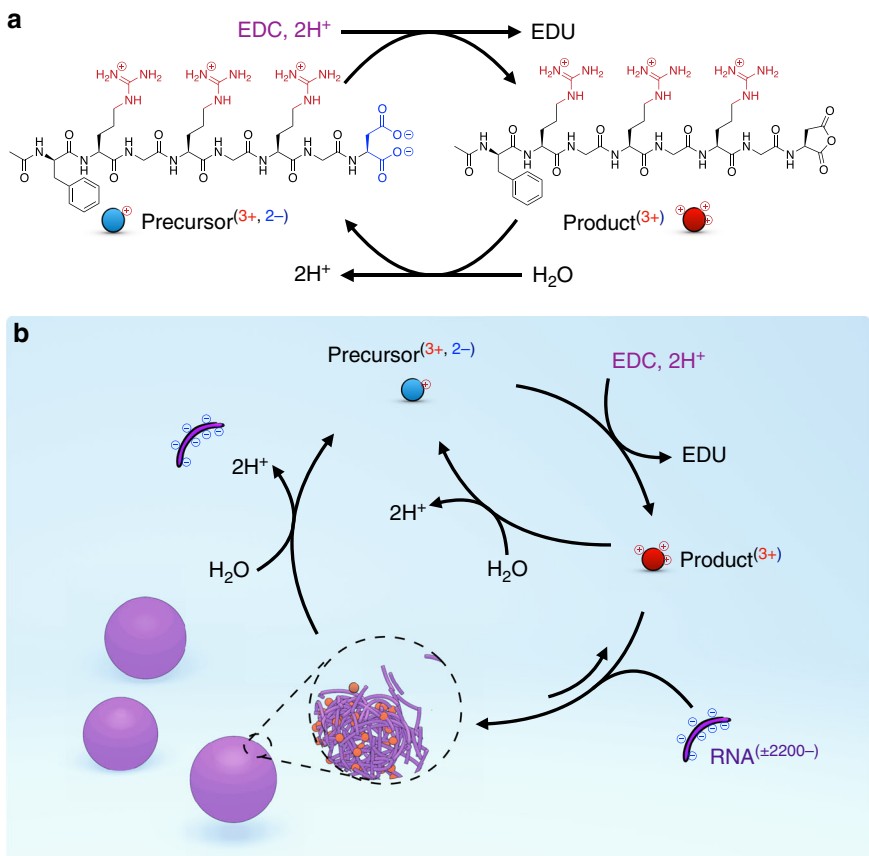

**Fig. 1 Coacervation between a peptide and RNA driven by a chemical fuel. a** The chemical reaction cycle that converts a chemical fuel (EDC) into waste (EDU) while removing two negative charges on a peptide (precursor) resulting in a cationic transient anhydride (product). **b** A schematic representation of the chemical reaction cycle combined with complex coacervation with RNA (poly-U). The influx and efflux of droplet materials are regulated by the chemical reaction cycle.

much slower than the kinetics of the reaction cycle allowing the droplets to persist for longer. We refer to these droplets as metastable droplets.

We performed the same experiments with various fuel concentrations to make a phase diagram (Fig. 2e and Supplementary Fig. 7). We found that more than 7.5 mM fuel was required to induce an increase in turbidity, which corresponds to 0.9 mM product. This result suggests a critical coacervation concentration of 0.9 mM anhydride product. Between 7.5 mM and 40 mM fuel, we found dynamic droplets. Above 40 mM of fuel, the droplets were metastable. We tested the effect of the RNA concentration on the droplet's behavior (Fig. 2f and Supplementary Figs. 8 and 9). For any RNA concentration, no increased turbidity was observed below 7.5 mM fuel, indicating that the RNA concentration was not the limiting factor for droplet formation. At lower RNA concentrations, less fuel was required to obtain the metastable droplets, e.g., 25 mM fuel or more for 1.4 mM RNA. We hypothesized that these metastable droplets can complex most RNA in the solution and that addition of more fuel results in more peptide product per RNA. The increased ratio of peptide to RNA results in an increased packing density compared to dynamic droplets, which may increase the energy barrier for disassembly. We thus measured the RNA concentration in the phase outside of the droplets. We used fluorescently labeled RNA to form dynamic droplets (25 mM fuel, 4.1 mM RNA), and, after 2 min, the droplet-containing solution was centrifuged to separate the two phases. Fluorescence spectroscopy showed that 10% of the initially added RNA remained in the supernatant (see Supplementary Fig. 10).

In contrast, no RNA was found in the supernatant when we did the same experiment with metastable droplets (60 mM fuel). The data from these experiments indicate that dynamic droplets leave behind some RNA in the outer phase, whereas metastable droplets capture all RNA.

Next, we measured the concentrations of fuel and product after filtering the droplet solution (see Supplementary Fig. 11). The filtration removed all droplets, and the measurements were thus indicative of the concentration of fuel and product outside of the droplets. Moreover, this data, combined with the dataset without filtration, was used to calculate the respective concentrations inside and outside of the droplet phase. For both dynamic (25 mM fuel) and metastable droplets (60 mM fuel), the concentration of fuel with or without filtration was similar, which suggest almost all fuel was present in the phase outside of the droplets at all times (see Supplementary Fig. 11a). This observation suggests that the activation of our reaction cycle predominantly occurs outside of the droplets in accordance with our design in Fig. 1. In contrast, we found a large decrease in the concentration of product when the samples were filtered compared to without filtration, especially early in the cycle (see Supplementary Fig. 11b). That observation suggests that the droplets comprise a significant amount of the anhydride product. Specifically, after 2 min, we measured that 1.7 mM product remained in the filtrate for the dynamic droplets (roughly 55% of the total product). Next, we measured the volume of the droplet phase by centrifuging the total solution and measuring the volume of the pellet. That volume, combined with the missing fraction of product, allows us to calculate the concentration of product inside

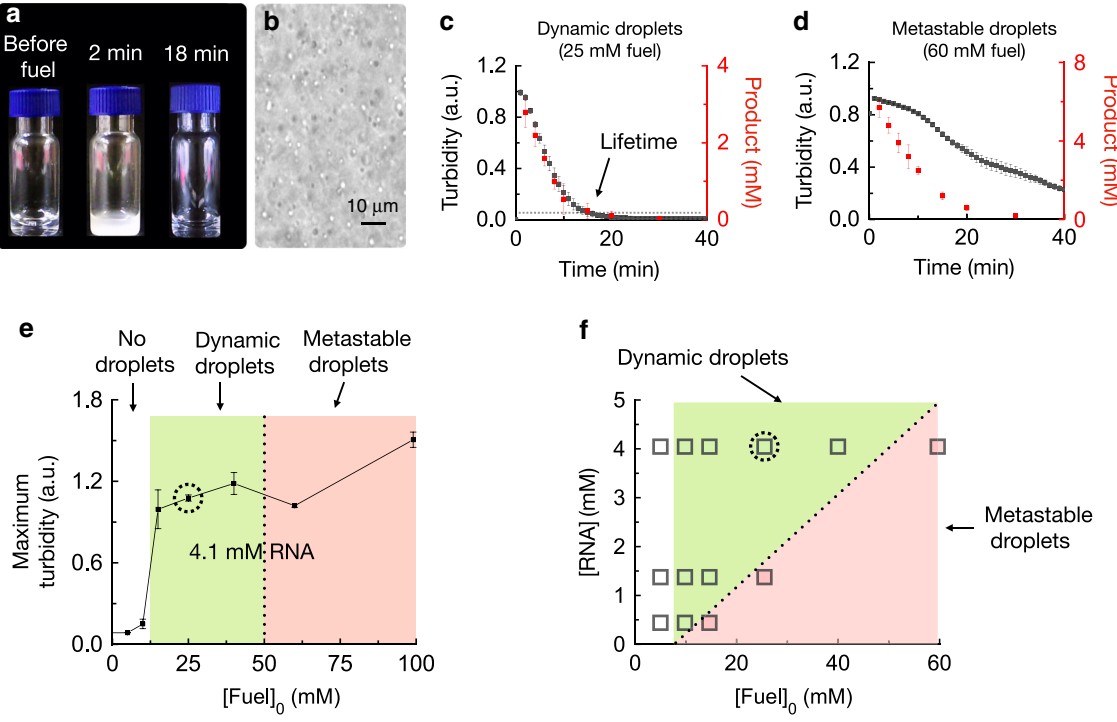

**Fig. 2 Macroscopic analysis of the droplets. a** Photographs of solutions of 23 mM Ac-FRGRGRGD-OH (precursor), 25 mM EDC (fuel), and 4.1 mM poly-U (RNA, monomer concentration), before, 2 min after and 18 min after the addition of fuel. **b** Bright field-micrograph of a solution described in **a** after 2 min. **c**, **d** Absorbance of 600 nm light as a measure of turbidity (left y-axis, black traces) and anhydride product concentration in the solution (right y-axis, red traces) as a function of time for the same condition as described in **a** with 25 mM fuel (**c**) or 60 mM fuel (**d**). Dynamic droplets show a strong correlation between the concentration of product and the turbidity. **e** The maximum turbidity as a function of fuel for samples with 4.1 mM RNA. The three regimes are shaded: no droplets (white), dynamic droplets (green), and metastable droplets (red). **f** Behavior of the droplets as a function of RNA monomer concentration and amount of fuel. All error bars show the standard deviation from the average (N = 3).

the droplets. We found that, after 2 min, ~0.9 M product was inside the dynamic droplets, while ~1.8 M was present inside the metastable droplets (see Supplementary Fig. 11c). These observations mean that metastable droplets comprised roughly two times more product than dynamic droplets. Considering that the amount of RNA in the droplet phase was almost similar in these experiments (*vide supra*), the ratio of product to RNA was significantly higher in metastable droplets. Specifically, the ratio of cations to anions, after 2 min, was 2.0 for the metastable droplets compared to 1.1 for the dynamic droplets.

From the experiments above we can conclude that precursor and RNA do not sufficiently interact, and cannot form droplets without activation of the peptide. The active product, in contrast has a higher affinity to the RNA leading to phase-separated droplets. Since coacervate droplets are typically highly hydrated, the product hydrolyzes back to the precursor inside of the droplets, decreasing the affinity between the peptide and RNA. In the case of dynamic droplets that loss in affinity leads to the immediate dissolution of droplets. In the case of metastable droplets, the deactivation of the product does not immediately disassemble the droplets. We hypothesize that, even though droplet dissolution is thermodynamically favored, the disassembly process is kinetically hindered. Compared to dynamic droplets, more peptide per unit of RNA is present in metastable droplets, which may result in an increased barrier of disassembly. We have observed such differences in the energy landscape in other chemically fueled assemblies[46,53].

**Droplet evolution and disassembly.** We further studied the emergence, evolution, and decay of the dynamic and metastable

droplets by confocal microscopy using Cy3-tagged A15 hybridized with RNA (Fig. 3a, b). In the first part of the cycle of dynamic droplets, the droplets were spherical and polydisperse (Fig. 3a). Halfway through the cycle, vacuoles (holes) started to appear in the remaining droplets. Towards the end of the cycle, the droplets suddenly fragmented in smaller, not perfectly spherical droplets. These final fragmented droplets finally dissolved. We quantified the above described trend by calculating the droplet number, their size, and their average fluorescence intensity (Fig. 3c–e, black traces). Within a minute after addition of 25 mM fuel, we observed a high number of droplets (Fig. 3a, c). In line with the turbidity measurements, the droplet number decayed after 3 min. Towards the end of the cycle, the number of droplets suddenly increased (Fig. 3c). The droplet volume immediately increased early in the cycle and started to decrease after 12 min (Fig. 3d). We found a high fluorescence intensity that rapidly decayed as time progressed (Fig. 3e). The decay in the intensity points to the fast release of RNA as droplets collapse.

When we performed the same experiments with the metastable droplets (60 mM fuel), we also found a high number of spherical, polydisperse droplets in the first minutes (Fig. 3b). Again, the droplet number rapidly decreased (Fig. 3c, red traces). However, droplets were still present even after complete fuel consumption (e.g., 60 min). After 20 min, the average volume of the metastable droplets remained constant (Fig. 3d). The most obvious difference between dynamic and metastable droplets was in their mean intensity of the fluorescent RNA (Fig. 3e). In dynamic droplets, the intensity decreased as the cycle progressed while in metastable droplets, it did not change significantly throughout the cycle. We conclude that the RNA remained in metastable droplets, long after all product had hydrolyzed (e.g., 40 min).

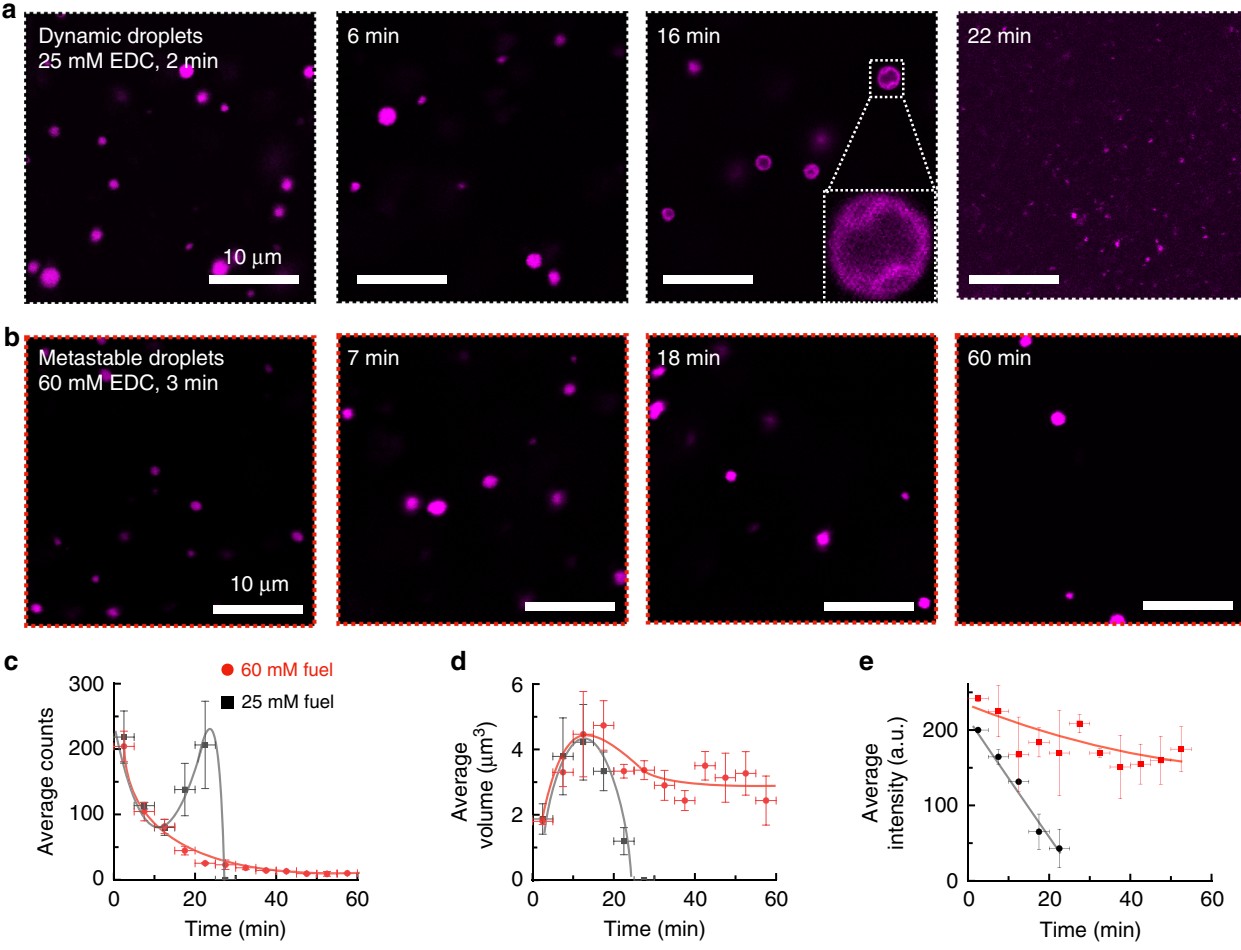

**Fig. 3 Microscopy analysis of dynamic and metastable droplets. a**, **b** Confocal micrographs of solutions of 23 mM precursor and 4.1 mM poly-U (Cy3-A15 hybridized) after the addition of 25 mM EDC (**a**) or 60 mM EDC (**b**). **c**–**e** Statistical analysis of confocal micrographs of the solutions described in **a** (black squares) and **b** (red circles) for the number (**c**), volume (**d**), and the average intensity (**e**) of the respective droplets after addition of fuel. Vertical error bars show the standard deviation from the average ($N = 3$). Horizontal error bars show binned data of 5 min. The lines are added to guide the eye. Source data are provided as a Source Data file.

In other words, the peptide is able to hold together the droplet to some degree after it has been deactivated.

Confocal microscopy time-lapses with higher acquisition rates revealed further dynamic behavior in the dynamic droplets (Fig. 4a and see Supplementary Movie 1). We could not observe the droplet nucleation, likely because it occurred rapidly after fuel addition (20 mM fuel). In the first minutes, we did observe a frequent fusion of the small droplets (Fig. 4b), which is common for coacervate-based droplets.[54] As time progressed, fusion became less apparent. After 12 min, we observed the first formation of vacuoles. In some cases, we found multiple vacuoles per droplet that grew or fused until they became the major component of the droplet (Fig. 4c). Interestingly, as the vacuoles grew, the droplet itself also increased in size, likely because of water uptake (see Supplementary Movie 2). We hypothesize that vacuoles are a result of deactivation of the peptide and the consequent efflux with RNA. The efflux is hindered by a diffusion barrier formed by the remaining part of the droplet, which yields in the accumulation of RNA at the droplet periphery. Moreover, studies of membraneless organelles in vitro[22] and in vivo[55] showed the dynamic formation of similar vacuoles. Around 14 min after fuel addition, we observed the first dissolution events of droplets (Fig. 4d, e). The droplet lost its roundness, and protrusions formed. Excitingly, these protrusions pinched off and became independent segments of the original droplets

(Fig. 4e and Supplementary Movies 3 and 4). In some cases, these segments drifted off by Brownian motion and could survive for tens of seconds before finally dissolving. Such droplet fragmentation would be a form of ill-controlled asymmetric division, which occurs as the original droplet is losing integrity due to the loss of anhydride product, which we will further study in future work.

From these combined experiments, we conclude that dynamic droplets almost immediately nucleate and take up RNA rapidly in response to fuel. Then, their average volume grows, while their number decreases due to fusion events. When product deactivation is faster than peptide activation, the product concentration starts decreasing, and the droplets release RNA, as can be concluded from the decline in the fluorescence intensity. In some cases, this release of RNA is hindered by the droplet itself and results in vacuole formation, i.e., areas with low RNA surrounded by a diffusion barrier with high RNA concentration. Towards the last minutes of the cycle, the droplets rapidly collapsed resulting in fragmentation and further release of the RNA.

**Transient up-concentration of functional RNA.** Complex coacervate droplets containing RNA have frequently been proposed as a potential protocell that emerged at the origin of life.[12,56] To demonstrate that our dynamic droplets can indeed up-concentrate functional RNA during their limited lifespan, we

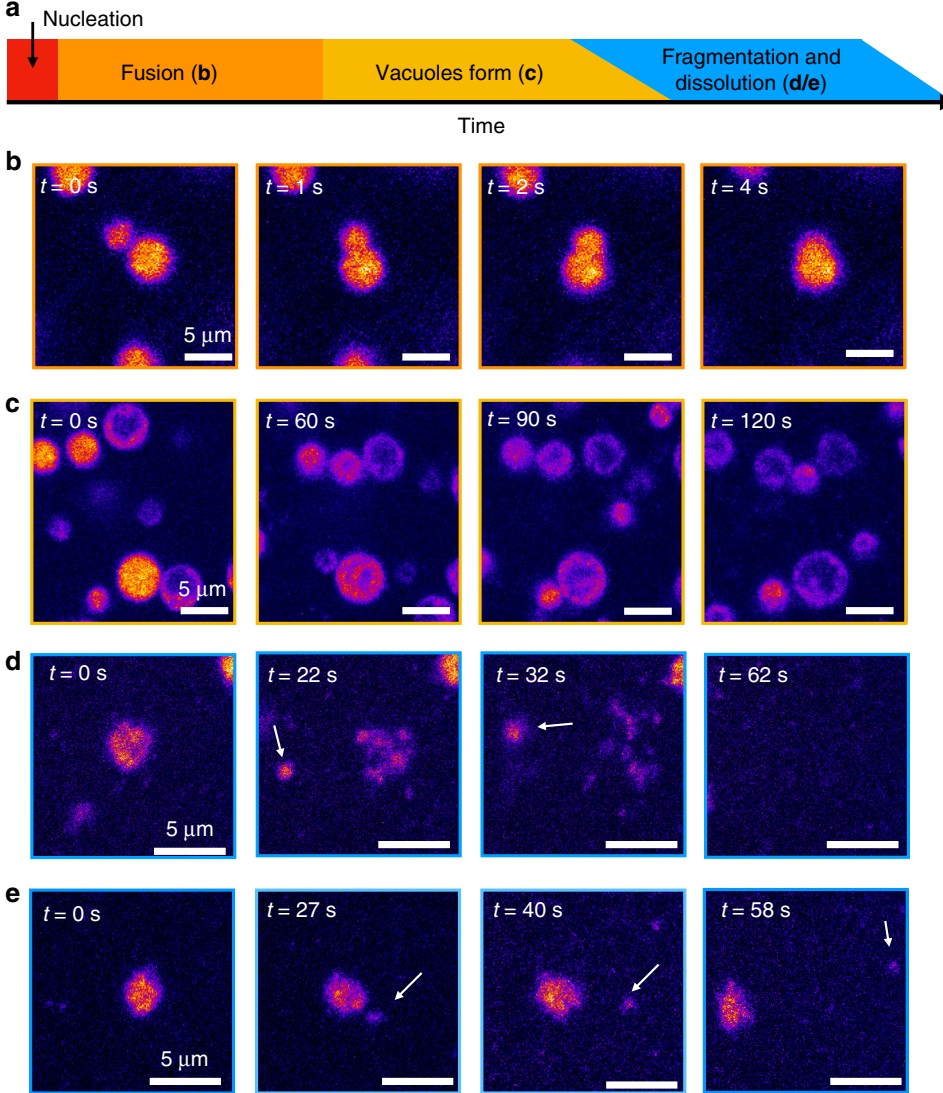

**Figure 4 Microscopy analysis of the behavior of dynamic droplets. a** The sequence of events occurring during the lifetime of dynamic droplets.
**b**–**e** Selected events of confocal micrograph time-lapses of solutions of 23 mM precursor and 4.1 mM poly-U (Cy3-A15 hybridized) after addition of 20 mM EDC. **b** Fusion of droplets at the beginning of the cycle (<5 min). **c** Vacuole formation towards the end of the cycle (>12 min). **d**, **e** The fragmentation and dissolution of a droplet, ~15 min after fuel addition. In some cases, the fragments survive for tens of seconds as indicated by the white arrows. Experiments were performed for $n > 10$.

investigated the evolution of partitioning of functional RNA sequences into the dynamic droplets (Fig. 5a–c). We used a ligating ribozyme, a cleaving ribozyme, and a fluorophore-binding aptamer (SunY,[57] Hammerhead,[12] and Broccoli,[58] respectively). Their sequence and synthesis details are provided in Tables 3–6. We added fuel to the solutions containing the precursor, poly-U, and one of the functional RNA sequences, and imaged the droplets by confocal microscopy. Five minutes after the addition of fuel (25 mM), we observed that all three functional RNA sequences had partitioning coefficients >10. (Fig. 5b and Table 3). In line with other reports, the partitioning seemed to be size-dependent; i.e., the longest sequence had the highest partitioning coefficient (SunY, 187 nucleotides, $K = 37$), while the shortest strand had the lowest (Hammerhead, 44 nucleotides, $K = 13$). As the cycle was progressing, an increasing amount of functional RNA was observed outside of the droplets until the droplets dissolved (Fig. 5c). Taken together, upon emergence, our droplets partition the functional RNA, and release it upon decay,

resulting in the transient up-concentration of functional RNA driven by a chemical reaction cycle.

To test whether functional RNA remains active in our droplets, we first tested the robustness of the reaction cycle against metal ions, such as potassium and magnesium, and temperature which are important biochemical parameters (see Supplementary Fig. 12). In general, the turbidity decreased with increasing salt concentration which is common for coacervation due to screening effects. Below 50 mM of salt as well as at 37 °C, fuel-driven droplets can still be formed and thus opening the possibility for compartmentalized biochemical reactions. We therefore examined the binding of the Broccoli aptamer to its ligand DFHB1T by fluorescence spectroscopy. DFHB1T is non-fluorescent, but, upon binding with Broccoli, emits bright green fluorescence.[58] When we added the ligand to a buffered solution of precursor, poly-U and Broccoli at low to medium levels of magnesium and potassium, the fluorescence intensity increased by a factor of 5 (Fig. 5d). Addition of 10 mM fuel decreased the intensity by 30% in the first minutes

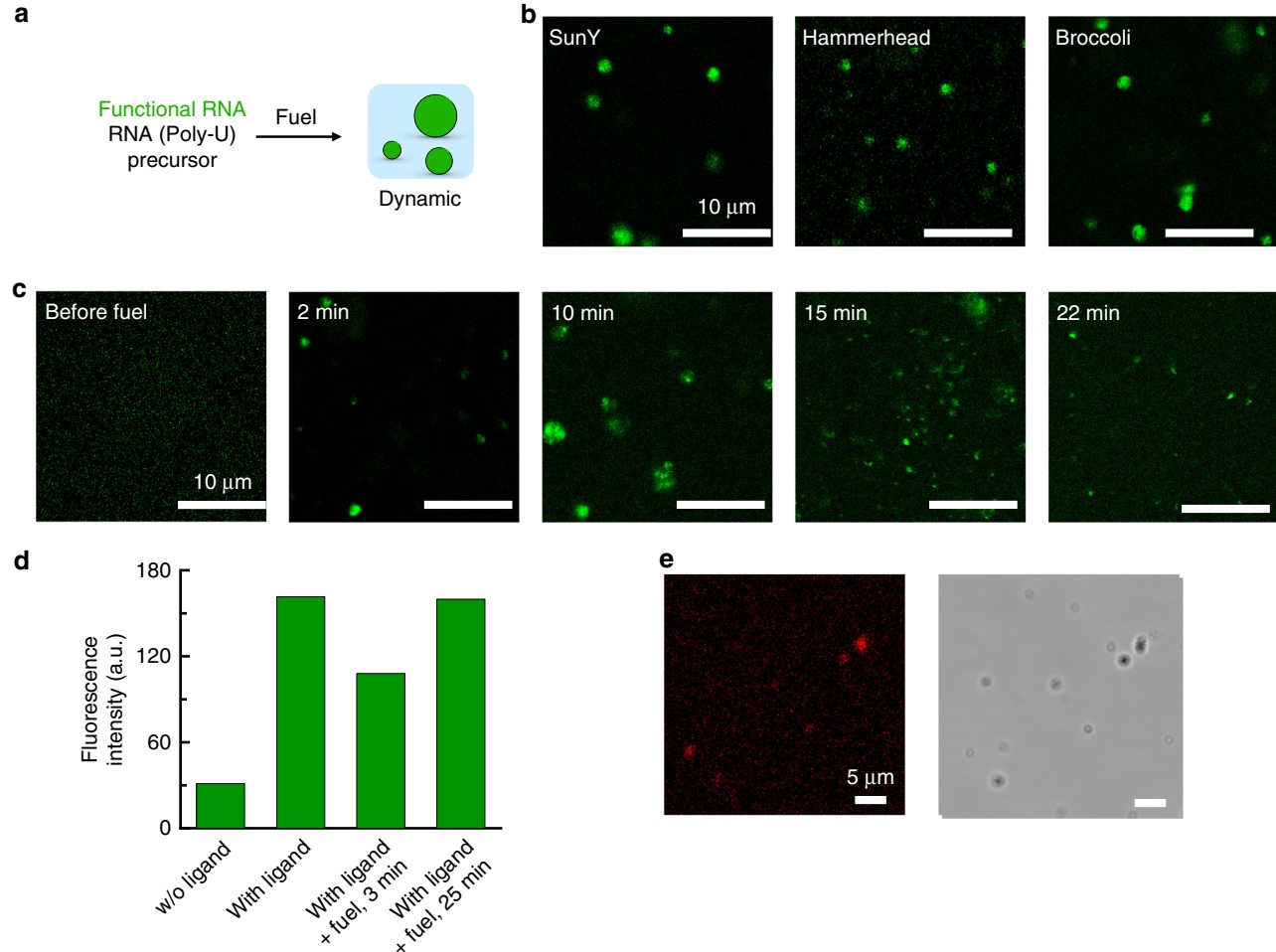

**Fig. 5 Functional RNA inside fuel-driven droplets. a** Schematic representation of the experimental procedure for dynamic droplets with functional RNA. **b** Confocal micrographs of 23 mM precursor, 4.1 mM poly-U, 25 mM EDC with 0.2 µM Cy5-RNA (SunY ribozyme, Hammerhead ribozyme or Broccoli aptamer), 5 min after addition of EDC. **c** Confocal micrographs of the SunY containing solution described in **d** (with 27 mM EDC) at different time points before or after EDC addition. **d** Fluorescence intensity of solutions containing the Broccoli aptamer with or without DFHB1T (ligand), in the presence and absence of droplets. Maximum fluorescence intensity at 504 nm. Standard conditions with 2 mM MgCl$_2$, 30 mM KCl, and 1.5 µM Broccoli aptamer. Addition of 10 mM EDC (fuel) to induce droplet formation. **e** Confocal and bright field micrographs under the conditions described in **d**, 5 min after the addition of 15 mM EDC. Experiments were perfomed for $n = 2$. Source data are provided as a Source Data file.

followed by a complete recovery once all fuel has been consumed. The decrease in intensity is likely because the droplets increase the sample's turbidity. We performed the same experiment at the confocal microscope to examine where the binding of the aptamer with the ligand occurs once droplets are present. In line with the broccoli partitioning, the broccoli-DFHB1T complex colocalizes within the droplets by more than an order of magnitude (Fig. 5e). We thus conclude that functional RNA can remain in its folded active state inside the droplets which is an important prerequisite for ribozyme activity. In future work, the combination with self-replicating RNA strands could result in the up-concentration and activation of catalytic RNA sequences whose activity helps to sustain the droplets.

We made droplet formation via complex coacervation a fuel-driven process. RNA-containing droplets emerge in response to fuel and decay when the fuel is depleted. We envision that our droplets could be powerful to study how chemical reactions can regulate liquid–liquid phase-separated droplet behavior, which could provide valuable insight in the mechanisms that govern membraneless organelles. We also believe these droplets can serve as a great model for protocells, as they spontaneously emerge and decay. We show that they transiently partition functional RNA

which is able to remain in its active folded state. More excitingly, we show the first signs of spontaneous asymmetric division towards the end of a coacervate droplet cycle.

## Methods

**Materials**. N,N′-Diisopropylcarbodiimide (DIC), ethyl (hydroxyimino)cyanoacetate (Oxyma), (Nova-biochem®),4-(dimethylamino)-pyridine (DMAP), Wang resin, protected amino acids (Fmoc-D(OtBu)-OH, Ac-F-OH, Fmoc-G-OH, Fmoc-R(Pbf)-OH, Fmoc-K(Boc)-OH, Fmoc-H(Trt)-OH), piperidine (99%), trifluoroacetic acid (99%, TFA), triisopropylsilane (TIPS), polyuridylic acid potassium salt (poly-U), 1-ethyl-3-(3-dimethylaminopropyl)carbodiimide (EDC), solvents (acetonitrile (ACN), N,N-dimethylformamide (DMF), dichloromethane (DCM), diethyl ether, and 4-morpholineethanesulfonic acid (MES) buffer were all purchased from Sigma-Aldrich and used without any further purification unless indicated otherwise. Nuclease-free water was freshly prepared by filtration (DURAN) of MQ-water. Cy3-A15 and Cy5-A15 were purchased from biomers.net GmbH. ssDNA oligonucleotides were purchased from IDT. The ribozymes (SunY, Hammerhead) and the Broccoli aptamer were synthesized by T7 polymerase run-off transcription (see Methods section). All peptides were synthesized using standard fluoren-9-ylmethoxycarbonyl (Fmoc) solid-phase peptide synthesis on Wang resin (see methods).

**Peptide synthesis and purification**. All peptides were synthesized via solid-phase synthesis on a CEM Liberty microwave-assisted peptide synthesizer. The purity of the peptide was analyzed by electrospray ionization mass spectrometry in positive mode (ESI-MS) as well as analytical HPLC.

*Primers, RNA, and splints.* All DNA primers were bought from IDT (see Supplementary Table 4). Primers were annealed and filled in using the Taq polymerase to create dsDNA templates used for T7 transcription and the RNA sequence listed in Supplementary Table 5. The underlined sequences are complementary to the Cy5-tagged pentamer, whereas the rest of the 10 nucleotides are complementary to the 3′ end of the respective RNA (see Supplementary Table 6).

**General sample preparation.** For most experiments, we used these standard conditions: 23 mM Ac-FRGRGRGD-OH (tri-trifluoroacetic acid salt), 4.1 mM poly-U (uridine units), 200 mM MES, pH 5.3, and 25 mM EDC (in case of dynamic droplets) or 60 mM EDC (metastable).

Stock solutions of the precursor (75 mM, tri-trifluoroacetic acid salt), MES buffer (500-1000 mM), poly-U (600-1000 kDA, ~2-15 μg/μL), and EDC (1–2 M) were prepared in nuclease-free water. The pH of the peptide and MES stock were adjusted to 5.3. Typically, stock solutions of EDC and poly-U as well as the resulting peptide-poly-U solution in MES were prepared freshly for each experiment. Reaction networks were started by addition of EDC to the peptide-poly-U solution in MES.

**ITC measurements.** All ITC measurements were performed using a MicroCal PEAQ-ITC instrument (Malvern Pananalytical) by injecting (26 points, 1.5 μL/inj) with a total of 37.5 μL of Ac-FRGRGRGD-OH (38 mM in 200 mM MES, pH 5.3) into 250 μL of poly-U (1.1 mM uridine units in 200 mM MES, pH 5.3) into the cell at 25 °C, reaching a final molar ratio of 6.6 (peptide/RNA). Using the MicroCal PEAQ-ITC Analysis software (v1.22), for each experiment first, a control experiment was subtracted, where the peptides (38 mM in 200 mM MES, pH5.3) were titrated into 200 mM MES (pH5.3) in the absence of RNA. All experiments were performed in triplicate ($N = 3$).

**Turbidity measurements and lifetime determination.** Turbidity measurements were carried out at 25 °C on a Microplate Spectrophotometer (Thermo Scientific Multiskan GO, Thermo Scientific SkanIt Software 6.0.1). Measurements were performed in a non-tissue culture treated 96-well plate (Falcon, flat bottom). Every minute, the absorbance of the 200 μL samples was measured at 600 nm. All experiments were performed in triplicate. The lifetime refers to the time it takes for the absorption to drop under 0.01 a.u. (blank subtraction) after EDC addition.

**Kinetic model.** A kinetic model written in MATLAB (v9.7.0) was used to predict the evolution of the EDC, precursor, and anhydride concentration over time. The model is described in detail in our previous work[41]. The rate constants we used in this work are given in Table S2. The Matlab-code we used is available here: https://github.com/BoekhovenLab/Dynamic-droplets. Briefly, a set of differential equations determines the concentration of EDC, precursor, O-acyl urea, EDU, and anhydride in the cycle every second.

**Determining the concentrations in the total reaction mixture.** The kinetics of the chemical reaction networks were monitored over time by means of analytical HPLC (Thermofisher Dionex Ultimate 3000, Hypersil Gold 250 × 4.6 mm, Chromeleon software 7.2 SR4). The turbid (in case of 5 mM EDC: clear) samples were directly injected without further dilution, and all compounds involved were separated using a linear gradient of ACN (2 to 98%) and water with 0.1% TFA.

Calibration was not possible for the anhydride product due to its intrinsic instability, proximity to the original precursor peak as well as low yield. Instead, an indirect quantification technique was used by converting the anhydride irreversibly into a mono-amide to determine the anhydride concentration (see Supplementary note 1). Measurements were performed at 25 °C. Briefly, after the addition of EDC (total volume = 200 μL) into a mixture of peptide and RNA in buffer (see general sample preparation), 20 μL of the turbid (in case of 5 mM EDC: clear) reaction mixture were added into 10 μL of an aqueous solution of benzylamine (900 mM) at each time point. The resulting clear solution (pH > 9) was then measured via HPLC to determine [mono-amide].

**Method to determine the anhydride and EDC concentrations outside the droplet phase.** Due to the inherent instability of the transient coacervate droplets, we chose a filtration protocol which separates the droplet phase from the phase outside of the droplets (see below). Moreover, unlike conventional centrifugation protocols in literature, this method allows for fast separation and enables us for fast quenching of the solution with benzylamine.[59] The speed of the separation method is important because anhydride product is constantly being produced in the supernatant due to the presence of EDC and precursor peptide, which can be calculated with our model. After calculating the anhydride concentration from the HPLC experiments of the filtrate, we corrected the resulting values for the anhydride product which was generated in the supernatant after separation from the droplet phase according to the model (typically it took 15–25 s between filtering and quenching). From these results, we can estimate the amount of EDC and anhydride in the droplet phase by subtraction from the total amount in both phases. In order to calculate the concentrations of

anhydride in the droplet phase, we measured the total droplet volume via centrifugation of the samples after fuel addition and comparing the droplet phase to a set of size standards visually.

After the addition of EDC (total volume = 100 μL) into a mixture of peptide and RNA in 300 mM buffer (see general sample preparation), the coacervate solution was filtered using regenerated cellulose filters (0.45 μm, 13 mm, Whatman SPARTAN) at each time point. In all, 20 μL of the resulting filtrate (clear solution) was added into 20 μL of an aqueous solution of benzylamine (1 M), while the residual 20–40 μL were added into a separate vial for EDC quantification. The anhydride (see benzylamine quenching method) and EDC concentrations of the supernatant were then calculated for both samples via HPLC. All experiments were performed in triplicate ($N = 3$).

**Fluorescence spectroscopy.** Fluorescence spectroscopy was performed on a Jasco (Jasco FP-8300, SpectraManager software 2.13) spectrofluorimeter with an external temperature control (Jasco MCB-100).

**Method to determine the RNA concentration in the supernatant.** In order to quantify the RNA inside the droplets, we hybridized poly-U with a Cy3-A15 (100 nM) prior to the experiment. After the addition of EDC (total volume = 300 μL) into a mixture of peptide and RNA in buffer (see general sample preparation), the coacervate solution was centrifuged 2 min after fuel addition at 20,412×g for 2 min. In total, 200 μL of the supernatant were pipetted into a cuvette and the fluorescence of the sample was measured between 550 and 600 nm. The experiment was performed for different fuel concentrations and we calculated the RNA concentration in the supernatant.

**Confocal fluorescence microscopy.** Confocal fluorescence microscopy was performed on a TCS SP5 II confocal microscope (Leica LAS AF, v2.6.3.8173) using a ×63 oil immersion objective. Samples (total volume = 15–50 μL) were freshly prepared as described above, but with 0.1–0.5 μM Cy3-A15 or Cy5-A15 as dye. In all, 5 μL of the sample were deposited on PEG-coated glass slides and covered with a 12-mm diameter coverslip. After tracking the droplets on the microscope for 4–5 min, again 5 μL of the sample was deposited on the glass slide and imaged instead of the previous sample portion. This process was repeated for the first 25 min of the experiment. Samples were excited with 561 nm and 633 nm laser and imaged at 570–620 nm and at 665–730 nm. Measurements were performed at 24 °C. Coating of glass slides consisted of a silanizing step with dichlorodimethylsilane following a PEGylation step with 1% Pluronic acid. All experiments were performed in triplicate ($N = 3$).

**Image analysis.** ImageJ's (2.0.0) preinstalled "analyse particles" package was used to analyze the number, the circumference, and fluorescence intensity of droplets under the assumption that the particles were perfectly spherical. The acquired data were then binned in 5-min bins in Excel (v16.16.24). Thus, each data point corresponds to five images (e.g., 1–5 min).

**Protocol for measuring the fluorescence intensity of the Broccoli-DFHB1T complex.** After the addition of 10 mM EDC (total volume = 150 μL) into a cuvette containing a mixture of peptide, poly-U, 1.5 μM Broccoli aptamer, 20 μM DFHB1T, 2 mM MgCl$_2$, and 30 mM KCl in MES buffer at standard conditions, the fluorescence of the sample was measured between 500 and 600 nm.

**Protocol for in vitro transcription of functional RNAs.** DNA templates for in vitro transcription were created by fill-in PCR using Taq-Polymerase (Promega). Briefly, two partially complementary ssDNA oligonucleotides were annealed and extended resulting in double-stranded DNA templates, which contained a 5′-terminal T7 promoter and the downstream transcription templates of interest. Following fill-in PCR, all dsDNA products were purified using the Monarch PCR & DNA Cleanup Kit (NEB). RNAs were synthesized by run-off transcription using the MEGAshortscript T7 Transcription Kit (Thermo Scientific). After column purification (Monarch RNA Cleanup Kit, NEB), RNAs were tagged with Cy5 fluorophore by ligating a RNA-pentamer modified with a 5′ phosphate and 3′ Cy5 using T4 RNA ligase 2 in the presence complementary DNA splint to the 3′-terminal 15 nucleotides of the RNA product. The ligated RNAs were then gel-purified after PAGE. The resulting RNA pellet after ethanol precipitation was suspended in ddH$_2$O and quantified on a Nanodrop measuring absorbance at 260 nm using their specific extinction coefficient calculated using OligoCalc (v3.27) (http://biotools.nubic.northwestern.edu/OligoCalc.html).

**Reporting summary.** Further information on research design is available in the Nature Research Reporting Summary linked to this article.

## Data availability
Source data are provided with this paper. Data is upon request from the corresponding author. Source data are provided with this paper.

## Code availability

The code used in the work is available here: https://github.com/BoekhovenLab/Dynamic-droplets.

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

## Acknowledgements

This project was funded by the Deutsche Forschungsgemeinschaft (DFG, German Research Foundation) – Project-ID 364653263 – TRR 235. J.B. is grateful for funding by the European Research Council (ERC starting grant, ActiDrops) under 852187, and the

Max Planck School Matter to Life supported by the German Federal Ministry of Education and Research (BMBF) in collaboration with the Max Planck Society. Mic.S. is grateful for funding by the DFG, German Research Foundation, SPP2191 Project number 402723784. Mar.S. is grateful for funding by the Deutsche Forschungsgemeinschaft (DFG) - Project number 411722921.

## Author contributions

J.B. and C.D. conceived the research and wrote the manuscript. J.B., C.D., M.T.S., Mic.S., and H.M. designed the experiments. C.D., F.S., Mar.S., B.A.K.K., F.S., M.T.S., H-S.K., E.S., and J.B. performed experiments and analyzed the data.

## Funding

## Competing interests

The authors declare no competing interests.
