## [Peer Review File · Nature Communications]

Reviewers' comments:

Reviewer #1 (Remarks to the Author):

In this manuscript, Boekhoven et al. show how a chemical fuel can be used to form transient supramolecular aggregates. The chemical fuel results in an increased charge on the peptide chain which as a consequence coacervates with RNA to form droplets. These droplets are metastable and hydrolysis of the peptide results in their disappearance. The authors show how these droplets can be kinetically trapped, by tuning the amount of chemical fuel. In addition, the authors study the uptake of material from the droplets surroundings.

The first part of the manuscript describes the versatile transient nature of the coacervates in a convincing and clear manner. One result in this section seems a bit puzzling though; Figure 3: When comparing the partition coefficient of the RNA with 60 mM fuel between 10 and 60 minutes, only a small decrease was observed. When comparing the average volume of the metastable droplets at these times, no significant change is observed, while their count drops by a factor of at least five. This large decrease in droplets would suggest that a large part of the RNA can no longer be present in the droplets. This seems inconsistent with the reported (rather constant) partition coefficient. How can the RNA remain in the droplets while the droplet number has diminished and their size remained constant?

In the introduction they imply that the product has the capacity to self assemble but the precursor does not (due to net charge differences), which is explained quite well, but when they come to describing the differences between the 'dynamic' and 'metastable' droplets in the text surrounding Figs 2c&d there is some rationale missing for why the turbidity doesn't decrease proportionally with the self assembled product concentrations obtained by HPLC. Perhaps I'm missing something about how these things work but I'm not entirely satisfied with just stating that the droplet (now containing no self assembled structures to provide droplet structure) is in a "kinetically arrested state" with high turbidity. Something seems off.

In the final part of the manuscript, the authors claim that the transient droplets take up components from their surroundings much faster than droplets in equilibrium. Their experimental support for this statement is however not convincing. Their experimental conditions studying the transient system allow for the formation of green droplets of which fusion with red droplets could result in the formation of orange droplets. Their control experiment (equilibrium conditions) does not allow for the formation of green droplets. Therefore fusion resulting in orange droplets could simply not occur (the control experiment is thus a very poor control, since the conditions are very different to the real experiment). No rapid uptake of RNA from the droplet's surroundings is required to explain their findings, simply droplet fusion would suffice. Although the authors mention that fusion alone is unlikely to cause the emergence of the orange droplets, they do not provide compelling evidence for this.

"Moreover, the controls (Figure S10) demonstrate that fusion alone cannot explain the rapid homogenization of dynamic droplets."

It appears that their control experiments of figure S10 offer no additional evidence (they are actually just the same pictures as Fig. 5, with the addition of the metastable droplets). The presence of some orange as well as completely green and red droplets at t=2 min instead of a gradual change from red and green to orange in fact suggests that uptake is dominated by droplet fusion instead of uptake from surroundings of the droplets.

A couple of additional comments which I think are important.

A lot of the language here is really over the top "a platform for synthetic life" - really? The last sentence of the conclusion is not intellectually supported in the manuscript "self-division combined with xx,xx, xx etc are all hallmarks of life,..." and I don't think the work presented here shown here, nice and interesting and (largely) publishable as it represents "a stepping stone in its

synthesis"

Most of the videos show something interesting going on but it's a stretch to call it promising signs of "self-division" or controlled splitting into daughter droplets. It looked much more like multiple, random sequential destruction events on the same droplet rather than a single continuous splitting process on any one of the droplets and I feel there's a distinction to be made, particularly if this is supposedly strengthening their "platform towards synthetic life" claim

Reviewer #2 (Remarks to the Author):

The authors create a system where RNA-uptaking droplets appear when there is fuel. These droplets can divide. The authors package this as steps towards synthetic life.

I think I have to take this paper at two levels. The first is the objective view of the work and the second is the subjective view of how interesting it is. From the former, it is solid work that is well-controlled and studied. The authors show dependency of droplet formation on RNA concentration and EDC "fuel" concentration; that droplets can be formed with a second batch of fuel; and they develop a phase diagram, which is a nice technical achievement. Confocal microscopy is used to visualize this cycle and discern the morphology of the system with so-called vacuole. Analogies are made to extant stress granules in terms of RNP composition and RNA exchange being accelerated by ATP depletion. Droplets are shown to take up three different ribozymes, but no activity assays are provided.

The second view is one of general interest. For me, I was not excited by this work. It came across as somewhat contrived. The authors should do a better job aligning this work with modern cells, but perhaps more importantly, give the reader an idea of whether this is robust or not. How tolerant is the system to variation in the components and the conditions. If it is a very fussy system, then it doesn't carry much weight in terms of possible models for life, or basis for synthetic life.

p6, l12. Do the authors know that these arginines form salt bridges with the RNA? Just because the can doesn't mean that they do.

The work would be more powerful if the ribozymes were shown to be active.

Reviewer #3 (Remarks to the Author):

This paper describes the behavior of complex coacervates formed from RNA and a short peptide, the charge on which can be controlled by the addition of EDC as a "fuel." This report is very exciting, particularly as the authors demonstrate the ability of their droplets to undergo cycles of formation and dissolution upon the addition of additional fuel. The novelty of this report is such that it warrants publication in Nature Communications, though I do have some questions related to their discussion (see below).

Specific Comments:

1. In the discussion at the top of page 10, the authors cite an experiment that compares the amount of product in the supernatant for conditions where dynamic vs. metastable droplets form. From just the text in the manuscript it is unclear what is meant by product. In looking at Figure S9, product appears to refer to the peptide-anhydride. This could be clarified. I would also suggest that since the take-home message from this section is that the anhydride is concentrated in metastable droplets, the authors could discuss the concentration of anhydride in the droplets,

rather than outside.

I also found it odd that the data in S9b only indicated the anhydride concentration for one time point, while the bar graph in S9c showed the relative concentration of anhydride for multiple times. Please include the raw data for the other plots in S9b.

2. While I agree that there is a correlation between larger quantities of anhydride peptide in the metastable droplets, compared with dynamic ones, what is the mechanism whereby this phenomenon would affect the dissolution of the droplets?
3. One assumption in these experiments is that the anhydride peptide formation is purely transient. However, is it possible that the transient nature of this species could be affected by the surrounding environment? For instance, in the metastable droplets, could it be possible that the peptides are slower to revert back to their less-charged state, and this is the reason for the slower decay? It should be possible to track this using a technique such as zeta potential or NMR, and such an understanding would add to the impact of the paper.
4. The authors use a series of control experiments where RNA of different colors was added to the sample to track the rate of droplet coalescence. They conclude that the rate of coalescence is insufficient to account for the colocalization of the two differently colored RNA in the droplets. The given explanation is that the flux of peptide in and out of the droplets must be resulting in rapid uptake/exchange of the RNA. While this explanation may be possible and is in agreement with the control experiments shown in Figure S10, I do not understand how the potential motion of a small peptide (which can diffuse quickly) should accelerate the diffusion of a much larger RNA molecule when a reaction is taking place. Were the two species to form a complex, wouldn't this complex diffuse even more slowly? I wonder if there an additional technique, such as fluorescence correlation spectroscopy (FCS) could be used with a mostly unlabeled sample to confirm the rapid motion of the RNA and peptide (this experiment would require labeling of the peptide and the RNA).
5. I would have liked to have known more about the peptides listed in the SI that were not observed to undergo reversible cycles of droplet formation and dissolution, and why precipitation was observed for those systems and not for the main one discussed in the manuscript.
6. The authors cite a number of papers that discuss coacervates in the context of membraneless organelles. However, there is a vast literature on complex coacervates from a number of significant groups that was not cited. I would suggest including a few more citations on coacervation generally. There have been a number of good review articles published in the last few years by the groups of Tirrell, Sing, Perry, and others.
7. In addition to references 18,19, I can recommend a paper by Semenov et al., Nature Chemistry (2015), <http://dx.doi.org/10.1038/nchem.2142>) which describes enzymatic control of coacervate droplets.
8. In Figure 5 it looks like there is a duplicate 10 μm label that ended up between the first two images. This shows up as a ghost/shadow in the image as the text is white.

Reviewers' comments:

Reviewer #1 (Remarks to the Author):

In this manuscript, Boekhoven et al. show how a chemical fuel can be used to form transient supramolecular aggregates. The chemical fuel results in an increased charge on the peptide chain which as a consequence coacervates with RNA to form droplets. These droplets are metastable and hydrolysis of the peptide results in their disappearance. The authors show how these droplets can be kinetically trapped, by tuning the amount of chemical fuel. In addition, the authors study the uptake of material from the droplets surroundings. The first part of the manuscript describes the versatile transient nature of the coacervates in a convincing and clear manner.

We thank the reviewer for the time invested in reviewing our work.

One result in this section seems a bit puzzling though; Partitioning coefficient metastable vs dynamic Figure 3: When comparing the partition coefficient of the RNA with 60 mM fuel between 10 and 60 minutes, only a small decrease was observed. When comparing the average volume of the metastable droplets at these times, no significant change is observed, while their count drops by a factor of at least five. This large decrease in droplets would suggest that a large part of the RNA can no longer be present in the droplets. This seems inconsistent with the reported (rather constant) partition coefficient. How can the RNA remain in the droplets while the droplet number has diminished and their size remained constant?

As the reviewer points out, there is a significant amount of RNA that is released by metastable droplets because they decay in the first 10 minutes. However, the RNA is diluted in the surrounding solution. Consequently, the loss of many droplets only marginally increases the concentration RNA in solution, whereas the droplets that "survive" did not decrease their intensity by much. That results in a minimal change in the partition factor. Whether correct or not, the partition factor in this context is confusing, and does not aid the reader. We thank the reviewer to point that out that.

Action: We exchanged the plot for one that shows the evolution of the mean droplet intensity and adjusted the manuscript accordingly.

In the introduction they imply that the product has the capacity to self-assemble but the precursor does not (due to net charge differences), which is explained quite well, but when they come to describing the differences between the 'dynamic' and 'metastable' droplets in the text surrounding Figs 2c&d there is some rationale missing for why the turbidity doesn't decrease proportionally with the self assembled product concentrations obtained by HPLC. Perhaps I'm missing something about how these things work but I'm not entirely satisfied with just stating that the droplet (now containing no self assembled structures to provide droplet structure) is in a "kinetically arrested state" with high turbidity. Something seems off.

Briefly, the precursor cannot induce droplet formation with the RNA without being activated first. However, in metastable droplets, towards the end of the cycle, droplets of exclusively precursor and RNA can exist. We explain the result by a pathway complexity phenomenon: inducing self-assembly occurs via a different pathway, then disassembly. Consequently, the energy barriers involved in these processes differ. We know the thermodynamic minimum of a solution of precursor and RNA is a state without droplets. However, that does not mean the solution will immediately reach that state after deactivation of the product. In other words, the energy barrier for disassembly of the precursor and RNA is relatively high. That means, that a droplet of which all the product has hydrolyzed is unstable from a thermodynamic point of view, and will, over time, disassemble. However, the energy barrier is in the range of the amount of thermal energy available, which means the droplet does not immediately disassemble. This metastability is something we encounter more often in chemically fueled assemblies (Nature Commun. 2017).

Somehow, the behavior is different for dynamic droplets: here we found that the deactivation and turbidity nicely correlated, which seems to suggest that the energy barrier for disassembly is lower than the available thermal energy. In other words, product that is deactivated leaves the droplets on the timescales of the experimental analysis. How do we explain the difference? The available thermal energy between the dynamic and metastable droplets is equal, so the difference must be in the energy barrier for disassembly, *i.e.*, metastable droplets have a higher barrier for disassembly compared to dynamic ones. Why? We can only offer a tentative mechanism. We know that the ratio product to RNA is higher in the metastable droplets. We thus assume that the increase ratio of peptide to RNA is responsible for a higher, denser packing which increases the energy barrier for disassembly. Briefly: a delayed disassembly profile is rather common in the field; being able to toggle between rapid disassembly or delayed disassembly is not.

Action points: in the main text, on page 10, we added further explanation for the reader.

In the final part of the manuscript, the authors claim that the transient droplets take up components from their surroundings much faster than droplets in equilibrium. Their experimental support for this statement is however not convincing. Their experimental conditions studying the transient system allow for the formation of green droplets of which fusion with red droplets could result in the formation of orange droplets. Their control experiment (equilibrium conditions) does not allow for the formation of green droplets. Therefore fusion resulting in orange droplets could simply not occur (the control experiment is thus a very poor control, since the conditions are very different to the real experiment). No rapid uptake of RNA from the droplet's surroundings is required to explain their findings, simply droplet fusion would suffice. *Although the authors mention that fusion alone is unlikely to cause the emergence of the orange droplets, they do not provide compelling evidence for this.*

"Moreover, the controls (Figure S10) demonstrate that fusion alone cannot explain the rapid homogenization of dynamic droplets."

It appears that their control experiments of figure S10 offer no additional evidence (they are actually just the same pictures as Fig. 5, with the addition of the metastable droplets). The presence of some orange as well as completely green and red droplets at t=2 min instead of a gradual change from red and green to orange in fact suggests that uptake is dominated by droplet fusion instead of uptake from surroundings of the droplets.

This comment is in line with the comment of reviewer three. We agree that we do not have enough evidence of the precise mechanism why dynamic droplets take up RNA faster. Currently, it looks like dynamic droplets are indeed fusing faster than metastable or static droplets. But, again, more data is required. In line with the comments of reviewer three, we removed this experiment from Figure 5, and the claims in the introduction and conclusion. Instead, we added an aptamer activity assay instead (see Reviewer two).

A couple of additional comments which i think are important.

A lot of the language here is really over the top "a platform for synthetic life" - really? The last sentence of the conclusion is not intellectually supported in the manuscript "self-division combined with xx,xx, xx etc are all hallmarks of life,..." and I don't think the work presented here shown here, nice and interesting and (largely) publishable as it is represents "a stepping stone in its synthesis"

We decreased the tone of the introduction and conclusions. We focused the abstract and conclusion on the role of these droplets as a model for membrane-less organelles. We adjusted the title accordingly.

Most of the videos show something interesting going on but it's a stretch to call it promising signs of "self-division" or controlled splitting into daughter droplets. It looked much more like multiple, random sequential destruction events on the same droplet rather than a single continuous splitting process on any one of the droplets and I feel there's a distinction to be made, particularly if this is supposedly strengthening their "platform towards synthetic life" claim

We obviously agree that this is not the clean self-division we see in extant cells. We also agree that the division we observe is a pathway towards dissolution, *i.e.*, as the droplet is losing its integrity, it is falling apart in smaller fragment. However, we were surprised to see that these smaller droplets remain present for a rather long time, and can thus be considered as individual droplet fragments (as opposed to fragments that are gone within seconds). Future work will tell if we can "save" the droplets and use them for evolution-based experiments. For now, we toned down the claims. We stopped referring to them as daughter droplets, and, as we have not shown yet that they are actually droplets, we refer to them as fragments.

Reviewer #2 (Remarks to the Author):

The authors create a system where RNA-uptaking droplets appear when there is fuel. These droplets can divide. The authors package this as steps towards synthetic life.

I think I have to take this paper at two levels. The first is the objective view of the work and the second is the subjective view of how interesting it is. From the former, it is solid work that is well-controlled and studied. The authors show dependency of droplet formation on RNA concentration and EDC “fuel” concentration; that droplets can be formed with a second batch of fuel; and they develop a phase diagram, which a nice technical achievement. Confocal microscopy is used to visualize this cycle and discern the morphology of the system with so-called vacuole. Analogies are made to extant stress granules in terms of RNP composition and RNA exchange being accelerated by ATP depletion. Droplets are shown to take up three different ribozymes, but no activity assays are provided.

We thank the reviewer for the time invested in reviewing our work. We appreciate the reviewer’s objective view.

The second view is one of general interest. For me, I was not excited by this work. It came across as somewhat contrived.

We are disappointed that the reviewer doesn’t share our enthusiasm about the system on a subjective level.

The authors should do a better job aligning this work with modern cells, but perhaps more importantly, give the reader an idea of whether this is robust or not. How tolerant is the system to variation in the components and the conditions. If it is a very fussy system, then it doesn’t carry much weight in terms of possible models for life, or basis for synthetic life.

We have carried out further experiments to probe its robustness or lack thereof. We now added data that demonstrate that the system is tolerant towards the addition of various cations important for ribozyme activity such as sodium, potassium and magnesium. We also show that the system works at both at 25°C and 37°C. As expected, with increasing temperature, the lifetime of the droplets decreases. Taking together, the screened conditions demonstrate the potential of our droplets of compartmentalizing biochemical reactions. We believe the peptide is reasonably robust towards varying conditions when it comes to chemically fueled droplet formation. Like other coacervate based systems, it will likely not tolerate very high salt concentration (>50 mM monovalent ions). See Figure S10 for the summarized results.

p6, l12. Do the authors know that these arginines form salt bridges with the RNA? Just because the can doesn’t mean that they do.

We agree with the reviewer that we did not demonstrate the presence of salt bridges. We thus removed that statement.

The work would be more powerful if the ribozymes were shown to be active.

We agree that the work would be more powerful. However, these experiments are substantial, and probably warrant their own study. We are working towards that, but realistically, especially considering the limited access to the labs given the current state of the affairs, results cannot be expected early next year at best.

However, what we have done is examined whether the aptamer is active inside and outside of the droplets. That is a good indication that the functional RNA remains folded and active in the presence of the reaction cycle and in the droplets. That, in turn, is an encouraging indication that ribozymes may be active in our droplets too. Thus, we tested the aptamer binding towards the ligand DFHB1T which emits green fluorescence after binding to the broccoli aptamer. After the addition of the ligand into a buffered solution containing precursor, poly-U and broccoli aptamer at low levels of magnesium and potassium, the fluorescence intensity increases by a factor of 5. Addition of 10 mM fuel leads to a decrease of fluorescence intensity by 30 % in the first minutes followed by a complete recovery once all fuel has been consumed. We performed the same experiment under a confocal microscope to examine where the binding of the aptamer with the ligand occurs once droplets are present. In line with the broccoli partitioning, the broccoli-DFHB1T complex colocalizes within the droplets by more than an order of magnitude. We thus conclude that functional RNA can remain in its folded active state inside the droplets which is a first requirement for ribozyme activity.

Reviewer #3 (Remarks to the Author):

This paper describes the behavior of complex coacervates formed from RNA and a short peptide, the charge on which can be controlled by the addition of EDC as a “fuel.” This report is very exciting, particularly as the authors demonstrate the ability of their droplets to undergo cycles of formation and dissolution upon the addition of additional fuel. The novelty of this report is such that it warrants publication in Nature Communications, though I do have some questions related to their discussion (see below).

We thank the reviewer for the time invested in reviewing our work, and appreciate his or her excitement.

Specific Comments:

1. In the discussion at the top of page 10, the authors cite an experiment that compares the amount of product in the supernatant for conditions where dynamic vs. metastable droplets form. From just the text in the manuscript it is unclear what is meant by product. In looking at Figure S9, product appears to refer to the peptide-anhydride. This could be clarified. I would also suggest that since the take-home message from this section is that the anhydride is concentrated in metastable droplets, the authors could discuss the concentration of anhydride in the droplets, rather than outside.

We adjusted the plot axis in S9b and S4 to “Anhydride product”. In general, that section was not well written. We rewrote it, and added another set of data. We now measured the total volume of the droplet phase. Combined with the concentrations of peptide outside of the droplets, we can calculate the exact concentration of product inside and outside of the droplets. Before, we were speaking in relative numbers.

I also found it odd that the data in S9b only indicated the anhydride concentration for one time point, while the bar graph in S9c showed the relative concentration of anhydride for multiple times. Please include the raw data for the other plots in S9b.

Graph S9c actually did contain three data points, but was confusing. In line with your comment above, we redid figure S9.

2. While I agree that there is a correlation between larger quantities of anhydride peptide in the metastable droplets, compared with dynamic ones, what is the mechanism whereby this phenomenon would affect the dissolution of the droplets?

Your comment is in line with the one of reviewer one. Briefly, we believe the increased ratio anhydride to RNA in the metastable droplets changes their packing density, which, in turn, results in a higher energy barrier for disassembly. The higher energy barrier means that disassembly is somewhat delayed after deactivation of the product.

Action point: We have better elaborated the difference between dynamic and metastable droplets in the main text.

3. One assumption in these experiments is that the anhydride peptide formation is purely transient. However, is it possible that the transient nature of this species could be affected by the surrounding environment? For instance, in the metastable droplets, could it be possible that the peptides are slower to revert back to their less-charged state, and this is the reason for the slower decay? It should be possible to track this using a technique such as zeta potential or NMR, and such an understanding would add to the impact of the paper.

We have tested the influence of droplet presence on the kinetics of the reaction cycle (Figure S4). We measure the concentration of anhydride and EDC by HPLC in the absence of droplets (by addition of less than 7.5 mM EDC), in the presence of dynamic droplets (by addition of 25 mM EDC), and in the presence of metastable droplets (by addition of 60 mM EDC). We then use a kinetic model that can predict the data. Importantly, we find no significant differences in the rate constants of activation or deactivation of the precursor and product, i.e., one set of rate constants can be used to describe the kinetics of all three possible phases (without droplets, metastable droplets, dynamic droplets). In other words, the presence or nature of the droplets does not affect the kinetics significantly and the environment does not affect the transient nature of the anhydride. We have further clarified that in the main text.

4. The authors use a series of control experiments where RNA of different colors was added to the sample to track the rate of droplet coalescence. They conclude that the rate of coalescence is insufficient to account for the colocalization of the two differently colored RNA in the droplets. The given explanation is that the flux of peptide in and out of the droplets must be resulting in rapid uptake/exchange of the RNA. While this explanation may be possible and is in agreement with the control experiments shown in Figure S10, I do not understand how the potential motion of a small peptide (which can diffuse quickly) should accelerate the diffusion of a much larger RNA molecule when a reaction is taking place. Were the two species to form a complex, wouldn't this complex

diffuse even more slowly? I wonder if there an additional technique, such as fluorescence correlation spectroscopy (FCS) could be used with a mostly unlabeled sample to confirm the rapid motion of the RNA and peptide (this experiment would require labeling of the peptide and the RNA).

We thank the reviewer for this point. Your comment forced us to have another close look at the data. By means of FRAP experiments, we aimed to test the increase uptake rate of RNA. But, the second dataset was not sufficient to confidently conclude an increased uptake. In fact, upon closer inspection, our assumption that droplet fusion was equal for all droplet-types was wrong. Dynamic droplets fuse significantly faster than metastable or passive ones. Perhaps that's due to the increased kinetics, but it is too early to tell. In line with the comments by reviewer one, we removed this experimental data set and its conclusion from Figure 5, and added an aptamer activity assay instead (see Reviewer two). Again, we thank the reviewer for being critical; we made a bad assumption resulting in a wrong conclusion.

5. I would have liked to have known more about the peptides listed in the SI that were not observed to undergo reversible cycles of droplet formation and dissolution, and why precipitation was observed for those systems and not for the main one discussed in the manuscript.

Our intention was to offer the list of peptides to the community who may want to follow up on the work, thus preventing that work is performed twice (it is a list of negative results). However, we did not analyze the precipitates in too much detail, but we know that they contain significant amount of both peptide and RNA. In the case of the Fmoc-containing peptides, it seemed that the hydrophobic and pi-pi contributions of the Fmoc group were too strong (even at low peptide concentration such as < 5mM), meaning that once the negative charges were removed via EDC, the formed peptide-RNA complex was too strong to be dissolved by the hydrolysis of the peptide, and the samples remained kinetically trapped in a precipitate. When we exchanged the Fmoc group for the Ac-F group, it reduced the RNA affinity and led to more dynamic assemblies. However, the spacing and amount of cationic charges were also crucial; as arginines closer to each other with a significant higher charge density produced more stable peptide-RNA complexes. It is interesting to note that these RG and FG motifs in our peptides are often found in proteins with intrinsically disordered regions which are the major components of membraneless organelles. Balancing the charge density is an important factor to consider, which has been already covered in many reviews, *e.g.*, the ones you suggested to cite.

Action points: we modified Table S1 with further explanations and added a note regarding the glycine spacing into the manuscript.

6. The authors cite a number of papers that discuss coacervates in the context of membraneless organelles. However, there is a vast literature on complex coacervates from a number of significant groups that was not cited. I would suggest including a few more citations on coacervation generally. There have been a number of good review articles published in the last few years by the groups of Tirrell, Sing, Perry, and others.

We thank the reviewer. We're relatively new to the field and have focused too much on the biological side of the field in the introduction. Action points: We included references of several other groups working in the field.

7. In addition to references 18,19, I can recommend a paper by Semenov et al., Nature Chemistry (2015), <http://dx.doi.org/10.1038/nchem.2142> which describes enzymatic control of coacervate droplets.

We love that work, but had completely forgotten that it also coupled the reaction cycles to droplet formation. Action points: the citation has been added.

8. In Figure 5 it looks like there is a duplicate 10 μ m label that ended up between the first two images. This shows up as a ghost/shadow in the image as the text is white.

Action point: We cleaned up the figure.

REVIEWER COMMENTS

Reviewer #1 (Remarks to the Author):

I have to admit that I didn't go through everything in the paper in detail, but appears to have changed in a good way - the most important criticism was the bit about the uptake from their surroundings, which has been removed and they seem to have addressed most of the other points as well.

Reviewer #2 (Remarks to the Author):

The authors have addressed my concerns. They have made steps towards showing the system is robust, as it tolerates a modest range of temperatures and cation identities, although not cation concentration. This strengthens the paper. There isn't demonstration of ribozyme activity, but there is now demonstration of aptamer activity and that helps to show an RNA can be functional. Overall, I can support publication at Nature Communications.

Reviewer #3 (Remarks to the Author):

I appreciate the effort and edits that the authors undertook with respect to this revised manuscript, and feel that it has strengthened the presentation. I have a minor comment/suggestion related to the possible mechanistic differences between the dynamic and metastable droplets, but am otherwise satisfied with the manuscript in its current state.

Specific Comments:

1. On page 11, the authors discuss a potential mechanism whereby the dynamic and metastable droplets behave differently. Of particular note, they suggest that the highly hydrated nature of the dynamic coacervate droplets helps to facilitate hydrolysis of the product back to the precursor. They suggest that the higher concentration of peptide per RNA in the metastable droplets might induce a "tighter packing" and therefore induce slower dissociation. I wonder if this "tighter packing" might also reduce the water content and thus the solvent exposure of the product peptide, thereby slowing the hydrolysis reaction. This might be a simpler explanation for the differences in behavior than a more non-specific discussion of "activation energy barriers."

Related to this, it was unclear to me whether or not the methods used to study the reaction kinetics would be sensitive to phenomena that were taking place in the droplets. My understanding is that the data were obtained by measuring the amount of the various species in the supernatant. However, the supernatant represents a significantly larger volume than that comprised by the droplets. Is it therefore possible that significant variations in the concentration of the various species in the droplets would not be captured in measurements of the supernatant due to dilution effects? Are there other ways that could be used to directly measure the rate of hydrolysis?

As a side comment, I found it somewhat frustrating that the reaction scheme for this model was not included in the documentation for this manuscript. Instead, it refers the reader to a different paper where the reaction scheme is written out in prose, rather than in reaction form. I can understand not going into the details of a previously developed model, but it would be straightforward to just write out the reaction scheme.

2. Page 11, line 6: the word "dissoltuion" is misspelled

REVIEWER COMMENTS

Reviewer #1 (Remarks to the Author):

I have to admit that I didn't go through everything in the paper in detail, but appears to have changed in a good way - the most important criticism was the bit about the uptake from their surroundings, which has been removed and they seem to have addressed most of the other points as well.

We thank the reviewer for the time invested in our manuscript.

Reviewer #2 (Remarks to the Author):

The authors have addressed my concerns. They have made steps towards showing the system is robust, as it tolerates a modest range of temperatures and cation identities, although not cation concentration. This strengthens the paper. There isn't demonstration of ribozyme activity, but there is now demonstration of aptamer activity and that helps to show an RNA can be functional. Overall, I can support publication at Nature Communications.

We appreciate your comments. The study on ribozyme activity will follow.

Reviewer #3 (Remarks to the Author):

I appreciate the effort and edits that the authors undertook with respect to this revised manuscript, and feel that it has strengthened the presentation. I have a minor comment/suggestion related to the possible mechanistic differences between the dynamic and metastable droplets, but am otherwise satisfied with the manuscript in its current state.

We appreciate the reviewer's input and clarify the questions below. We have updated the manuscript accordingly.

Specific Comments:

1. On page 11, the authors discuss a potential mechanism whereby the dynamic and metastable droplets behave differently. Of particular note, they suggest that the highly hydrated nature of the dynamic coacervate droplets helps to facilitate hydrolysis of the product back to the precursor. They suggest that the higher concentration of peptide per RNA in the metastable droplets might induce a "tighter packing" and therefore induce slower dissociation. I wonder if this "tighter packing" might also reduce the water content and thus the solvent exposure of the product peptide, thereby slowing the hydrolysis reaction. This might be a simpler explanation for the differences in behavior than a more non-specific discussion of "activation energy barriers."

While it is a simpler explanation, we have two pieces of evidence that the hydrolysis rate is not playing a major role here. The evidence is that in three cases (without droplets, with dynamic droplets, and with metastable droplets), the concentration profiles can be modelled with one set of rate constants (Figure S4). Therefore, the kinetics have to be similar within the different droplets and in solution (within a certain error margin, of course). Secondly, the anhydride in the metastable droplets is already hydrolyzed after 20 minutes (Figure 2d, as evidenced by HPLC on the total solution), however the droplets are still present (as evidence by turbidity and microscopy). Therefore, the precursor must somehow stabilize these droplets and not the anhydride (it's simply not there anymore).

Action points: we have toned down the claim in the main text and stopped using "tighter packing" for which we indeed have no evidence.

It is worth mentioning, we have seen exactly the behavior you describe in oil droplets (DOI: 10.1038/s41467-018-04488-y). The difference in behavior of the kinetics with and without droplets is, in that case, immediately obvious.

Related to this, it was unclear to me whether or not the methods used to study the reaction kinetics would be sensitive to phenomena that were taking place in the droplets. My understanding is that the data were obtained by measuring the amount of the various species in the supernatant. However, the supernatant represents a significantly larger volume than that comprised by the droplets. Is it therefore possible that significant variations in the concentration of the various species in the droplets would not be captured in measurements of the supernatant due to dilution effects? Are there other ways that could be used to directly the rate of hydrolysis?

We measured the concentrations in the total phase and in the supernatant phase. We calculate the concentrations in the droplet phase by: $[x]_{\text{total phase}} - [x]_{\text{supernatant phase}}$.

We measured the total concentration by quenching the solution with benzylamine. That immediately dissolves the droplets and traps the anhydride. We do the same with the supernatant phase, but first filter out the droplets.

Action points: we clarified that we injected turbid solutions into the HPLC in the SI. Furthermore, we have added a reaction scheme for the benzylamine quench in the SI.

As a side comment, I found it somewhat frustrating that the reaction scheme for this model was not included in the documentation for this manuscript. Instead, it refers the reader to a different paper where the reaction scheme is written out in prose, rather than in reaction form. I can understand not going into the details of a previously developed model, but it would be straightforward to just write out the reaction scheme.

We agree. The full cycle and description of the each step is included in the SI.

2. Page 11, line 6: the word "dissoltuion" is misspelled

Corrected.

REVIEWERS' COMMENTS:

Reviewer #3 (Remarks to the Author):

I appreciate the effort and edits that the authors undertook with respect to this revised manuscript and look forward to its acceptance and publication.